# The Effect of Phonomyography Prototype for Intraoperative Neuromuscular Monitoring: A Preliminary Study

**DOI:** 10.3390/bioengineering11050486

**Published:** 2024-05-14

**Authors:** Yanjie Dong, Weichao Guo, Yi Yang, Qian Li

**Affiliations:** 1Department of Anesthesiology, West China Hospital, Sichuan University, Chengdu 610041, China; 2020224025501@stu.scu.edu.cn; 2Meta Robotics Institute, Shanghai Jiao Tong University, Shanghai 200240, China; guoweichao90@gmail.com; 3Department of Anesthesiology, Sichuan Provincial People’s Hospital, Chengdu 610072, China; 15108434044@163.com

**Keywords:** neuromuscular monitoring, phonomyography, acceleromyography, muscle sound

## Abstract

Quantitative neuromuscular monitoring, as extolled by clinical guidelines, is advocated to circumvent the complications associated with neuromuscular blockers (NMBs), such as residual neuromuscular block (rNMB). Nonetheless, the worldwide utilization of such methods remains undesirable. Phonomyography (PMG) boasts the advantages of convenience, stability, and multi-muscle recording which may be a promising monitoring method. The purpose of this preliminary study is conducting a feasibility analysis and an effectiveness evaluation of a PMG prototype under general anesthesia. A prospective observational preliminary study was conducted. Twenty-five adults who had undergone none-cardiac elective surgery were enrolled. The PMG prototype and TOF-Watch SX simultaneously recorded the pharmacodynamic properties of single bolus rocuronium at the ipsilateral adductor pollicis for each patient. For the primary outcome, the time duration to 0.9 TOF ratio of the two devices reached no statistical significance (*p* > 0.05). For secondary outcomes, the multi-temporal neuromuscular-monitoring measurements between the two devices also reached no statistical significance (*p* > 0.05). What is more, both the Spearman’s and Pearson’s correlation tests revealed a strong correlation across all monitoring periods between the PMG prototype and TOF-Watch SX. Additionally, Bland–Altman plots demonstrated a good agreement between the two devices. Thus, the PMG prototype was a feasible, secure, and effective neuromuscular-monitoring technique during general anesthesia and was interchangeable with TOF-Watch SX.

## 1. Introduction

Residual neuromuscular block (rNMB), with an incidence of 64.7% in the United States, is a common postoperative complication related to neuromuscular blockers (NMBs) after general anesthesia which may weaken the strength of respiratory and pharyngeal muscles, increasing the incidence of postoperative pulmonary complications [1,2,3]. Recently, both the American Society of Anesthesiologists (ASA) and the European Society of Anaesthesiology and Intensive Care (ESAC) updated their recommendations which call for quantitative neuromuscular-monitoring methods to be utilized to avoid catastrophic outcomes such as rNMB related to NMBs [4,5]. However, the global adoption and clinical implementation of quantitative neuromuscular monitoring remain suboptimal [6,7,8].

Multifactorial causes resulted in the global neuromuscular-monitoring dilemma. On the one hand, shortcomings of traditional quantitative neuromuscular-monitoring techniques are pivotal in contributing to the aforementioned phenomena. Existing traditional quantitative neuromuscular-monitoring techniques need elaborate setup, are susceptible to artifacts, or lack of alternative monitoring of muscles [9,10]. On the other hand, though plenty of novel neuromuscular-monitoring devices continue to cut a striking figure, their feasibility remains elusive [11,12,13]. And TOF-Watch SX (Organon Ltd., Dublin, Ireland), a traditional acceleromyography (AMG)-based neuromuscular-monitoring equipment item, continues to be the most utilized apparatus both in scientific research and in clinical practice [14]. 

A wide range of microphones are extensively applied for capturing biological signals of hemodynamics, respiratory mechanics, and joint movement [15]. Phonomyography (PMG), also known as acoustic myography or muscle sound, is a technique that uses a microphone to detect low frequency acoustic waves triggered by muscle fiber movement [16,17,18]. Previous studies have already proved the feasibility of this technique for intraoperative neuromuscular monitoring [19,20]. Owing to its advantages like convenience, stability, and multi-muscle recording, PMG might be a superior neuromuscular-monitoring method, and possesses a border clinical prospect [9]. Yet, no PMG-based equipment could be widely accessible in clinical routines. And whether the effect of PMG on neuromuscular monitoring during different blocking degrees is comparable to or exceeds that of TOF-Watch SX at the adductor pollicis muscle remains elusive.

Our previous work identified that PMG is equipped with the ability of muscular assessment in chronic muscle weakness and pathological muscles (Appendix A), and thus developed a PMG prototype for intraoperative neuromuscular monitoring [21,22,23]. The aim of this preliminary study is comparing the effect of the PMG prototype and TOF-Watch SX to reflect the pharmacodynamic properties of single bolus rocuronium during general anesthesia, finishing a feasibility analysis and effectiveness evaluation for this PMG prototype, laying a foundation for device update and further study, bridging the aforementioned research gap, and advancing the implementation of neuromuscular monitoring.

## 2. Materials and Methods

This preliminary prospective observational study was approved by the Ethics Committee of West China Hospital of Sichuan University on 8 July 2022 (2022-978). The study was registered in the Chinese Clinical Trial Registry (ChiCTR2200067117). The study was conducted at West China Hospital of Sichuan University, and all participants provided written informed consent according to the Declaration of Helsinki. The study process is illustrated in Figure 1.

### 2.1. Patients

A total of 25 patients aged 18 to 65 years with ASA physical status ASA I–III, body mass index (BMI) 18.5 to 24.9 kg/m^2^, who underwent none-cardiac elective surgery were enrolled. 

Exclusion criteria included the following: neuromuscular diseases or family history of myasthenia gravis, history of hand injury, patients who were administrated drugs that may influence the function of neuromuscular junctions, patients who were allergic to drugs in this study, and patients with latent difficult mask ventilation or difficult endotracheal intubation [24]. All inclusion and exclusion criteria were determined based on the Good Clinical Practice Guideline and previous studies [11,14].

### 2.2. Perioperative Anesthetic Management

The anesthesia-management strategy for each participant was standardized. Anesthesia-induction medications included midazolam 1–2 mg, sufentinel 0.2–0.3 ug/kg, propofol 1.5–2.5 mg/kg, and rocuronium 0.6 mg/kg. Perioperative basic monitoring contained electrocardiogram, non-invasive blood pressure monitoring (blood pressure cuff was placed on the opposite side of neuromuscular monitoring), and pulse oxygen saturation monitoring.

General anesthesia was maintained with intravenous propofol 4–12 mg/(kg·h) combined with remifentanil 0.05–0.3 ug/(kg·min). The allowable range of blood pressure fluctuation was within ± 20% of baseline. Appropriate sedation and analgesia were maintained for stable intraoperative blood pressure. Hypotension was treated with 0.2 mg metaraminol, 1.5–3 mg ephedrine or a fluid bolus, according to clinical routine. During the experiment, no additional NMBs except the induction bolus were administered. Flurbiprofen for multimodal analgesia and tropisetron for prevention of postoperative nausea and vomiting were also routinely administered. During the anesthesia recovery period, no NMB antagonists were utilized.

### 2.3. Neuromuscular Monitoring

The adductor pollicis muscle of the left hand was the target muscle; whether that is the dominant hand would not affect the effectiveness of neuromuscular monitoring for both PMG and AMG [25,26]. The baseline body temperature of the aforementioned forearm was recorded to ensure the temperature was higher than 32 °C (before the patient entered the operating room, a hot water bag would keep the temperature of the palm of the experimental side ≥ 32 °C). The left hand of patients would be secured to an arm board with tape. And the placement of a TOF-Watch SX (TOF-Watch SX 2.5.INT; Organon Ltd., Dublin, Ireland) strictly followed the requirement of Good Clinical Practice Guideline for neuromuscular blocking agents [14]. First of all, abrasion and cleaning of the skin was completed before the two surface Ag/Cl electrodes were placed over the ulnar nerve of the experimental arm. The negative electrode was placed near the wrist and another electrode was 3–4 cm away. Second, no preload but an acceleration sensor was placed on the study thumb and the PMG (version 1.0) microphone would be attached to the midpoint of the thenar eminence of the study hand, see Figure 2. Throughout this study, the PMG prototype only serviced as a muscle sound receiver and saver, all electric stimulation was emitted from the TOF-Watch SX to ensure patients’ safety and to avoid iatrogenic injury. The characteristics and design of the PMG prototype for signal recording, processing, and extracting can be found in our previous studies [21,22,23].

After midazolam, sufentinel and propofol were administered and the patient was unconscious; the PMG prototype and TOF-Watch SX were switched on. The TOF-Watch SX then for 5 s at 50 Hz performed limb stimulation to shorten the calibration time. Next, the CAL-2 algorithm of TOF-Watch was selected [27]. A consecutive four muscle twitch, namely T1, T2, T3, and T4, was incurred via TOF (four consecutive stimulations with a frequency of 2 Hz, a pulse width of 0.2 ms, and a stimulus cycle of 15 s). The TOF ratio (TOFr), referred to as T1/T4, displayed at this moment was considered as baseline data. The TOFr during recovery would be standardized through dividing it by the last baseline TOFr. If three consecutive TOFr values fluctuate by more than 100 ± 5%, the calibration was considered to have failed and the position of the acceleration sensor or the hand posture should be adjusted, and the calibration should be repeated [28]. After successful calibration, 0.6 mg/kg of rocuronium was administered intravenously within 5 s. At the same time, the TOF-Watch SX was adjusted to emit a 2 Hz TOF stimulus every 15 s. When the TOF-Watch SX indicated a zero TOF count or the TOFr was no longer decreasing, endotracheal intubation was performed immediately. 

Intraoperatively, the depth of muscle relaxation was continuously monitored via the PMG prototype and TOF-Watch SX. TOF stimulation parameters were the same as during the anesthesia-induction period. When the TOF Watch SX showed a TOFr of 0, Post-Transient-Count (PTC) Stimulation was initiated at 3 min intervals. PTC is a stimulation model that refers to 16 consecutive single twitch stimulations with 2 Hz and a pulse width of 0.2 ms after a 50 Hz tetanic stimulation. If PTC was ≥1 and TOF count = 0, a state of deep neuromuscular blockade (dNMB) state had been achieved. During this period, PTC stimulation was maintained until T1 recovered, which means a moderate neuromuscular blockade (mNMB) state emerged. During the moderate neuromuscular blockade period, TOF stimulation was resumed every 15 s. The endpoint of observation was reached when the TOFr of the TOF-Watch SX reached 0.9 or above on three consecutive occasions or when the operation was completed. Tracheal extubation was conducted by a senior anesthesiologist in the operation room or post-anesthesia care unit (PACU).

### 2.4. Study Outcomes

The primary observational outcome was recovery time/T90% (the time from the administration of rocuronium to the recovery of TOFr to 90% of standard value of the two equipment items, respectively), universally known as 0.9 TOFr time. 

The secondary observational outcomes were as follows: Tonset, known as onset time (the time from the administration of rocuronium to T1 was suppressed to 95%T1 of the two equipment items, respectively);T0 (the time from the administration of rocuronium to dNMB);Tdeep, known as dNMB time;T1 (the time from the administration of rocuronium to mNMB);Tmoderate, known as mNMB time;T25%, namely 25% T1 time (the time from the administration of rocuronium to the recovery of T1 to 25% of the standard value of the two equipment items, respectively);T75%, also known as 75% TOFr time (the time from the administration of rocuronium to the recovery of the TOFr to 75% of the standard value of the two equipment items).

### 2.5. Data Analysis

The data were statistically analyzed using R 4.3.2 (R Foundation for Statistical Computing) and Microsoft, Excel 2016. All data were tested for normality using the Shapiro Wilk test. Paired *t*-test or Wilcoxon signed-rank test were used to compare the pharmacodynamic data of rocuronium collected from the two apparatuses. Pearson’s correlation analysis and Spearman’s correlation analysis were applied to assess the association between data of the two devices during different time points of general anesthesia. The difference in measurements between the two devices was analyzed using the Bland–Altman plots. Limits of agreement (LOA) were calculated as the means of differences ±1.96 × SD of the two equipment items, resulting in 95% CI lower and upper limits. A *p* < 0.05 was considered statistically significant in this study.

## 3. Results

Twenty-one patients were finally included in this study. The clinical characteristics of the included patients are shown in Table 1. Three patients were excluded due to technical problems with the PMG prototype. The mean age of enrolled patients was 40 years. The average BMI of the included patients was 22.58 kg/m^2^ and both patients’ ASA classification was class II. The surgical procedure included otolaryngology surgery (*n* = 15), general surgery (*n* = 6), and neurosurgery (*n* = 1) with a mean anesthesia time of 147.32 min and an average surgical time of 99.77 min. Of the 176 planned pair data, 45 pairs were missing owing to destabilization of the PMG prototype or TOF-Watch SX.

Data are displayed as mean ± SD with paired *t*-test or median (IQR) with Wilcoxon signed-rank test.

As shown in Table 2, in terms of primary observational outcome, the median T90% recorded by the PMG prototype was 74.42 (57.98, 91.51) minutes, while TOF-Watch SX recorded 66.57 (58.14, 93.48) minutes of median T90% (*p* > 0.05). On secondary observational outcomes, the median Tonset recorded via the PMG prototype was 74.00 (60.00, 75.00) seconds, while TOF-Watch SX recorded 60.00 (60.00, 82.00) seconds of median Tonset (*p* > 0.05). The mean T0 was 9.38 ± 4.20 min for the PMG prototype and 8.48 ± 4.60 for TOF-Watch SX (*p* > 0.05). And the mean Tdeep was 20.42 ± 12.08 min for the PMG prototype and 22.74 ± 9.06 min for TOF-Watch SX (*p* > 0.05). In addition, the median T1 collected via the PMG prototype was 28.75 (25.47, 36.60) minutes, while another device collected 30.90 (24.93, 34,43) minutes of median T1 (*p* > 0.05). The median Tmoderate collected via the PMG prototype was 14.50 (8.75, 36.81) minutes, while another device collected 14.52 (10.52, 32.88) minutes of median Tmoderate (*p* > 0.05). Further, the median T25% was 35.76 (29.73, 47.81) minutes for the PMG prototype and 37.83 (33.31, 50.31) minutes for TOF-Watch SX (*p* > 0.05). The median T75% recorded via the PMG prototype was 65.82 (51.99, 82.71) minutes, while TOF-Watch SX recorded 56.27 (51.02, 87.43) minutes of median T75% (*p* > 0.05). 

The Spearman’s correlation test, displayed at Figure 3a, illustrated that a strong positive relationship between data recorded via the PMG prototype and TOF-Watch SX existed for outcomes of Tonset (r = 0.75, *p* < 0.001), T1 (r = 0.89, *p* < 0.010), Tmoderate (r = 0.81, *p* < 0.001), T25% (r = 0.93, *p* < 0.001), T75% (r = 0.94, *p* < 0.001), and T90% (r = 0.93, *p* < 0.001). In addition, the Pearson’s correlation test, presented at Figure 3b, also demonstrated that a strong positive relationship between data recorded by the two devices existed for outcomes of T0 (r = 0.91, *p* < 0.001) and Tdeep (r = 0.77, *p* < 0.01).

Bias with limits of agreement between measurements of the PMG prototype and TOF-Watch SX during onset, dNMB, mNMB, and recovery time are presented in Table 3. For onset time, as illustrated in Table 3 and Figure 4a, the bias between the two devices over the range of assessments was estimated to be −1.57 (95% CI, −8.34~5.20). The 95% LOAs were −30.72 to 27.58 (95% CI of lower LOA, −42.48~−18.96; 95% CI of upper LOA, 15.82~39.33). In addition, for dNMB time, as illustrated in Table 3 and Figure 4b, the bias between the two devices over the range of assessments was estimated to be 2.32 (95% CI, 2.32 (−3.24~7.87). The 95% LOAs were −12.91 to 17.54 (95% CI of lower LOA, −22.74~−3.07; 95% CI of upper LOA, 7.70~27.38). Furthermore, for mNMB time with data shown in Table 3 and Figure 4c, the bias between the two devices over the range of assessments was estimated to be −1.85 (95% CI, −5.18~1.49). The 95% LOAs were −15.00 to 11.30 (95% CI of lower LOA, −20.81~−9.19; 95% CI of upper LOA, 5.49~17.11). Finally, for recovery time with data displayed in Table 3 and Figure 4d, the bias between the two devices over the range of assessments was estimated to be 0.23 (95% CI, −1.91~2.36). The 95% LOAs were −14.956 to 15.41 (95% CI of lower LOA, −18.63~−11.28; 95% CI of upper LOA, 11.74~19.08). The above results indicate that the PMG prototype and TOF-Watch SX measurements were in good agreement.

## 4. Discussion

This preliminary study yielded preliminary results on the clinical efficacy of the PMG prototype. By comparing the pharmacodynamic data of a single dose rocuronium recorded via the PMG prototype with those recorded via the TOF-Watch SX, it was confirmed that the PMG prototype can be used to measure the onset, dNMB duration, mNMB duration, and myorelaxation recovery time of 0.6 mg/kg rocuronium-induced skeletal muscular paralysis in a clinical setting. The correlation between the two devices was strong, with no significant difference in the multi-temporal neuromuscular-monitoring data, indicating that this PMG prototype device has the potential to monitor neuromuscular function during anesthesia. 

Muscle sound was firstly discovered by Grimaldi [29]. As an objective reflection of mechanical contraction of skeletal muscle fiber, PMG has been widely applied in the fields of prosthetic limb control and chronic muscle disease screening [30,31,32]. However, only a few studies have reported assembled PMG-based devices in perioperative neuromuscular monitoring [33,34]. In previous studies, the minimal diameter of the acoustic microphone was 1.6 cm and the Root Mean Square (RMS) of the acoustic signal was the only parameter to represent muscle contraction intensity [9]. By contrast, the hardware composition and software settings of the PMG prototype were derived from our previous results in prosthetic control, which were significantly updated and optimized compared to the signal acquisition device in previous studies [21,22,23]. For acoustic signal gathering, the PMG prototype recruited a 6 mm diameter condenser microphone combined with an air chamber, which improved the quality of the acoustic signal while shielding against external noise. For data analysis, the PMG prototype was not only equipped with the capacity to integrate RMS of muscle sound and display real-time sound signals, but was also programmed to store all neuromuscular-monitoring information that could be reviewed retrospectively. 

Accurately identifying the onset time of NMB could help anesthesia providers to choose the optimal timing of endotracheal intubation, reducing the incidence of adverse events such as barking cough, airway injury, and difficult intubation [5,35]. Meistelman et al. [36] found that the onset time of rocuronium at the laryngeal muscle was faster than that at the adductor pollicis muscle. Therefore, the onset time of rocuronium is able to be reflected by monitoring the adductor pollicis muscle safely. Dascalu et al. compared the neuromuscular-monitoring effects of PMG and AMG at the adductor pollicis muscle of 12 patients who were undergoing elective surgery through an air chamber microphone [37]. They found that the correlation coefficient of the T1 ratio between the two devices before and after the administration of a nondepolarizing neuromuscular blocker during induction of anesthesia was 0.906, suggesting that the PMG and the AMG were equally effective in determining the onset time of a nondepolarizing neuromuscular blocker. Hemmerling et al. compared the neuromuscular-monitoring effect of PMG and AMG at the corrugator supercilii muscle in 20 patients undergoing general anesthesia, and they found that PMG recorded a longer onset time than that of AMG [38]. Our study focused on the adductor pollicis muscle and draws the result that the median onset time recorded via the PMG prototype was 74 s, while the median onset time recorded via TOF-Watch SX was 60 s. There was no significant difference in onset time between the two devices and the agreement was good. This result seems to be consistent with that of previous clinical studies targeting the adductor pollicis muscle, but different from those conducted at the corrugator supercilii muscle. One cause may be the different features of the target muscles. The corrugator supercilii muscle has fewer motor units and less muscle mass than the adductor pollicis muscle; thus, muscle contraction cannot cause displacement of the acceleration sensor but muscle sound still existed when maximal effect of rocuronium was looming, resulting in a shorter onset time for AMG. However, on the one hand, a difference in the tracheal intubation time of more than ten seconds is clinically meaningful enough, and on the other hand, a non-parametric test may be underpowered. Studies with larger sample sizes and more sophisticated PMG-based apparatuses are necessary. What is more, others have reported that the corrugator muscles are more sensitive than the adductor pollicis muscle in reflecting the onset time of NMBs in the laryngeal muscles [39]. The evaluation of the PMG prototype in the corrugator supercilii muscle will be clinically important in the future. For now, as the adductor pollicis muscle is recommended for neuromuscular monitoring in clinical guidelines and is the most frequently used site for neuromuscular monitoring in clinical settings, results from our study have sufficient universality and promotional value [4].

dNMB has gained increasing attention in the academic line due to its capability to improve surgical fields, to reduce occasional body movement, and to alleviate postoperative pain in certain surgical procedures [5]. Dhoneur et al. [39] proved that the adductor pollicis muscle is a reliable muscle for monitoring dNMB and the muscle relaxation recovery of the adductor pollicis during dNMB is closely related to the early paralysis recovery of the diaphragm in the human body. Zero literature pays attention to the effect of PMG during dNMB hitherto. Our study is the first report of the aforementioned issue. The mean duration of dNMB was 20.42 min for the PMG prototype and 22.74 min for TOF-Watch SX, with no significant difference. A 2.32 min shorter dNMB period could be explained by the sensitive features of PMG with respect to detecting subtle muscle strength recovery compared with AMG. Nevertheless, data of dNMB duration were limited and exhibited fluctuations with insufficient statistical power. Further large-scale studies focusing on dNMB surgeries are needed to confirm these findings. In any case, this result indicates that PMG could be utilized for dNMB monitoring, filling the research gap of PMG in this degree of muscle relaxation. Furthermore, it suggests the PMG prototype may hold the potential to be a reliable and sensitive neuromuscular-monitoring method for dNMB dependent surgery such as bariatric surgery, aneurysm surgery, and laryngeal microsurgery [40,41,42]. Yet, its efficacy and whether it could improve clinical outcomes of dNMB still need further investigation.

Recovery of muscle relaxation is the priority of intraoperative neuromuscular monitoring [43]. Incorrect monitoring information would cause an ignorance of rNMB and wrong timing for NMB antagonist administration. A TOFr recovered to 90% is the diagnostic cut point of rNMB [4]. Reflection of tussis and sufficient tidal volume are the characteristics of desirable muscle relaxation recovery. And the full recovery of the strength of respiratory muscles such as the laryngeal and diaphragmatic muscles is a prerequisite for the aforesaid features. Research has shown that the muscle-relaxation recovery time of the adductor pollicis muscle is longer than that of the laryngeal and diaphragmatic muscles, illustrating that the adductor pollicis muscle is an ideal peripheral target muscle for monitoring muscle relaxation recovery [44]. Dascalu et al. [37] compared PMG and AMG during anesthesia recovery at the adductor pollicis muscle after NMB antagonist administration in eight patients under general anesthesia. The results showed that when muscle relaxant was completely antagonized, the T1 recorded via AMG could recover to 84.4% of the baseline, while the T1 value recorded via PMG could recover to 97.1% of the baseline. However, Hemmerling et al. [38] compared the effect of PMG and AMG during anesthesia recovery at the corrugator supercilii muscle, with PMG having an earlier muscle-relaxation recovery time. In our study, the multiple time points of muscle-relaxation recovery (T25%, T75%, and T90%) recorded via the PMG prototype and TOF-Watch SX were approximately similar. Lacking T90% data and differences in target muscle may be the contrast of results between our study and that of Hemmerling’s [38]. Previous studies on the corrugator supercilii muscle offered an alternative choice for neuromuscular monitoring. The discrepancy of results between the corrugator supercilii muscle and adductor pollicis muscle, probably owing to the motor units and muscle mass characteristics of the corrugator supercilii muscle, suggests that the choice of target muscle can impact the measurement of neuromuscular blockade. Given that the adductor pollicis muscle is currently considered the most optimal site for neuromuscular monitoring, we believe results of our study hold certain clinical potential. Yet, it should be noted that AMG frequently overestimates the recovery time of the neuromuscular blockade. Therefore, the results need further investigation and validation through a comparison between PMG with other neuromuscular-monitoring methods such as electromyography (EMG).

It is obvious that limitations exist in this study. In the first place, this study was originally designed as a small-sample observational study for preliminary feasibility validation and subsequent technical optimization of this PMG prototype for neuromuscular monitoring. Consequently, the sample size is insufficient for comprehensively validating the efficacy of this equipment. A more well-designed study with a larger sample size is still necessary to achieve robust results. In the second place, rocuronium was the only NMB studied; prudent consideration should be taken while utilizing these results in clinical practice with other nondepolarizing NMBs and depolarizing NMBs. In the third place, although the neuromuscular-monitoring effects of the PMG prototype in different depths of muscle relaxation were discussed, there is a lack of data on the neuromuscular-monitoring effects of the PMG prototype in different populations such as critically ill, obese, geriatric, or pediatric patients. Last but not the least, given that the primitiveness of the PMG prototype and aging of the TOF-Watch SX, as it is no longer manufactured, hitches occurred frequently during this study.

Future works will be focused on enhancing both the PMG prototype itself and clinical trial designs. For algorithms, the integration of time domain and frequency domain analyses of PMG, employing both RMS and Power Spectral Density correlation coefficients, is planned to characterize muscle sound properties and to provide individualized neuromuscular monitoring. Additionally, the automatic extraction of spectral energy from PMG signals related to stimulation frequency will be applied to represent intraoperative muscle fiber activity. This feature may provide insights into patients’ short-term and long-term prognoses. For hardware, we aim to develop a combined microphone and stimulation electrode to enhance device integration and to create a lighter and more portable PMG device. The exploration of sensor fusion techniques, such as integrating Surface Electromyography (sEMG), to enhance the dimensions and precision of neuromuscular monitoring would also be initiated. Moreover, we would conduct randomized controlled trials with expanded sample size to investigate the effects of PMG in neuromuscular monitoring under different NMBs, NMB antagonists, and in diverse patient populations. Comparative studies will also be conducted between PMG and other quantitative neuromuscular-monitoring devices, such as EMG, to further validate the feasibility of PMG for neuromuscular monitoring.

## 5. Conclusions

This observational preliminary study demonstrated good agreement between the PMG prototype and TOF-Watch SX on onset time, dNMB time, mNMB time, and recovery time during 0.6 mg/kg of rocuronium-induced muscle paralysis at the adductor pollicis muscle under general anesthesia. Our findings suggested the feasibility and potential of the PMG prototype as a neuromuscular-monitoring tool during general anesthesia.

## Figures and Tables

**Figure 1 bioengineering-11-00486-f001:**
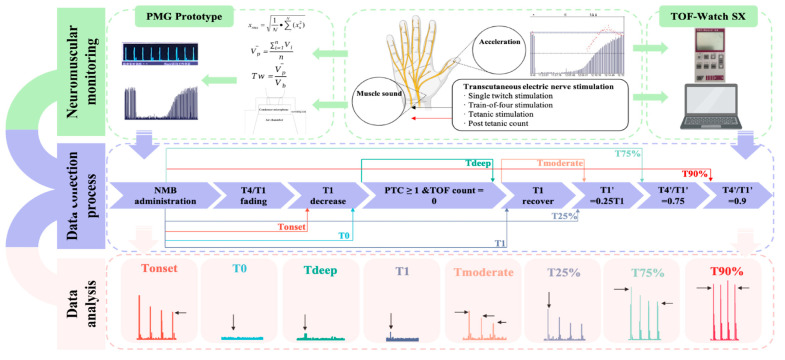
Design and operation process of this study.

**Figure 2 bioengineering-11-00486-f002:**
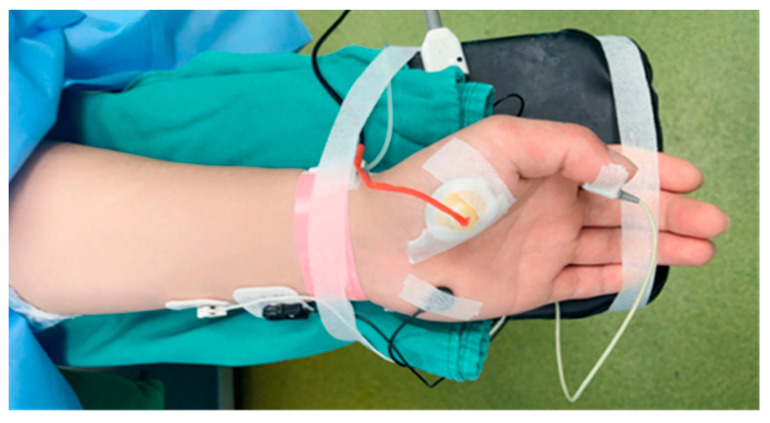
Placement of electrodes on the ulnar nerve and acceleration sensor of TOF-Watch SX on the thumb and acoustic sensor of PMG prototype on the thenar eminence with adhesive tape.

**Figure 3 bioengineering-11-00486-f003:**
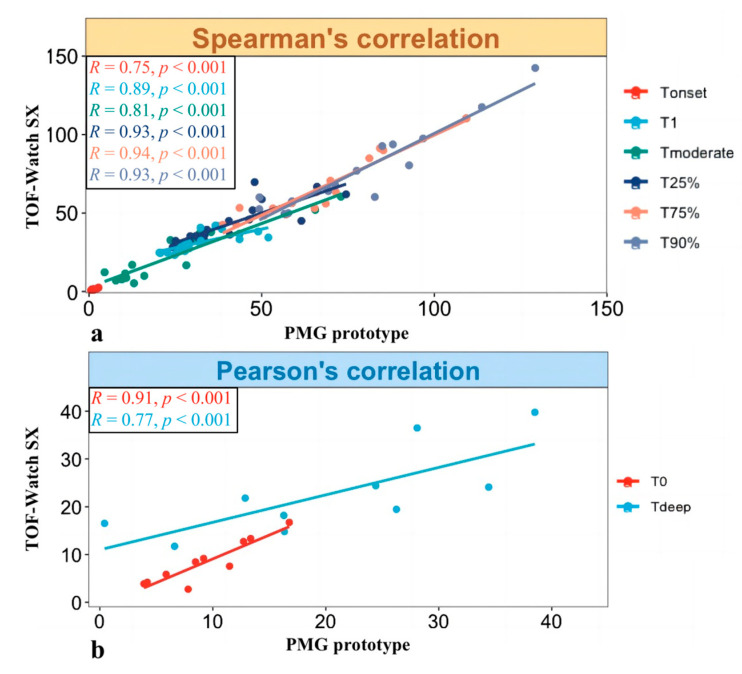
Correlation analysis of data of different outcomes recorded via the PMG prototype and TOF-Watch SX. (**a**)The Spearman’s correlation test. (**b**)The Pearson’s correlation test.

**Figure 4 bioengineering-11-00486-f004:**
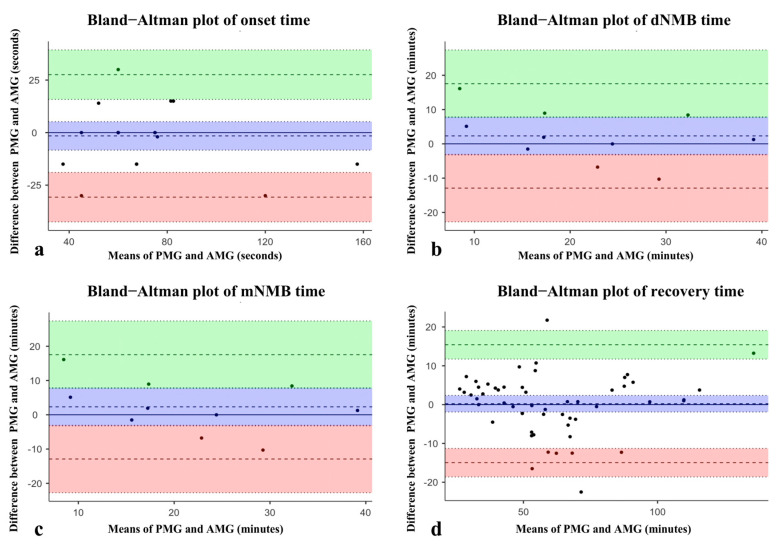
Bland–Altman plots of data for PMG prototype and TOF-Watch SX. Black solid lines and blue zones in the middle represent bias and its 95% CI. Black dotted upper lines and green zones represent upper limits of agreement and their 95% CI. Black dotted lower lines and red zones represent lower limits of agreement and their 95% CI. Black solid circles indicate a pair of data of a single patient. (**a**) Bland–Altman plot of onset time with only one pair of data outside of LOA. (**b**) Bland–Altman plot of dNMB time with all data inside of LOA. (**c**) Bland–Altman plot of mNMB time with all data inside of LOA. (**d**) Bland–Altman plot of recovery time with only two pairs of data outside of LOA.

**Table 1 bioengineering-11-00486-t001:** Clinical characteristics of included patients.

Variable	*N* = 22
Age (y)	40 ± 11.51
M/F	7/15
BMI (kg/m^2^)	22.58 ± 1.67
ASA (I/II/III)	0/22/0
Otolaryngology surgery	15
General surgery	6
Neurosurgery	1
Anesthesia duration	147.32 ± 55.82
Surgical duration	99.77 ± 44.74

y, years old; M, male; F, female; BMI, body mass index; ASA, American Society of Anesthesiologists. Data are displayed as mean ± SD.

**Table 2 bioengineering-11-00486-t002:** Outcomes recorded via PMG prototype and TOF-Watch SX.

Outcome	PMG Prototype	TOF-Watch SX	*p*-Value
Tonset (seconds)	74.00 (60.00, 75.00)	60.00 (60.00, 82.00)	0.520
T0 (minutes)	9.38 ± 4.20	8.48 ± 4.60	0.170
Tdeep (minutes)	20.42 ± 12.08	22.74 ± 9.06	0.370
T1 (minutes)	28.75 (25.47, 36.60)	30.90 (24.93, 34,43)	0.520
Tmoderate (minutes)	14.50 (8.75, 36.81)	14.52 (10.52, 32.88)	0.349
T25% (minutes)	35.76 (29.73, 47.81)	37.83 (33.31, 50.31)	0.059
T75% (minutes)	65.82 (51.99, 82.71)	56.27 (51.02, 87.43)	0.795
T90% (minutes)	74.42 (57.98, 91.51)	66.57 (58.14, 93.48)	0.679

**Table 3 bioengineering-11-00486-t003:** Bias and 95% limits of agreement between two devices for different durations.

Time Duration	Bias and 95% CI	95% Limits of Agreement	95% CI, Lower Limit of Agreement	95% CI, Upper Limit of Agreement
Onset time	−1.57 (−8.34~5.20)	−30.72 to 27.58	−42.48 to −18.96	15.82 to 39.33
dNMB time	2.32 (−3.24~7.87)	−12.91 to 17.54	−22.74 to −3.07	7.70 to 27.38
mNMB time	−1.85 (−5.18~1.49)	−15.00 to 11.30	−20.81 to −9.19	5.49 to 17.11
Recovery time	0.23 (−1.91~2.36)	−14.956 to 15.41	−18.63 to −11.28	11.74 to 19.08

Onset time, duration of Tonset; dNMB time, duration of Tdeep; mNMB time, duration of Tmoderate; recovery time, duration of (T25% + T75% + T90%).

## Data Availability

The data is not publicly accessible for the privacy of the research participants.

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
