# Peer review of "The Effect of Phonomyography Prototype for Intraoperative Neuromuscular Monitoring: A Preliminary Study"

_bioengineering, 2024, doi:10.3390/bioengineering11050486_

Round 1

Reviewer 1 Report

Comments and Suggestions for Authors

The interesting and observational study was well conducted and documented from an electromyographic and anaesthesiologic point of view. The adductor inch muscle of the left hand was the correct target muscle. It is difficult to understand the monitoring of muscle activity during anaesthetic administration.

Strengths

Scientific rigor for patient selection

Correct statistical analysis

The illustrated figure is very well described in detail.

Points of weakness

The number of patients is insufficient to establish the validity of the proposed monitoring techniques of patients is insufficient to establish the validity of the proposed monitoring technique

Author Response

Dear reviewer,

Thank you very much for your recognition of our work. With your help, we have made revisions to the manuscript, the details of which are as follows :

Point 1: The number of patients is insufficient to establish the validity of the proposed monitoring techniques of patients is insufficient to establish the validity of the proposed monitoring technique

Response 1: Thank you for the valuable comment and highlighting the limitation in our study. We believe that an insufficient sample size is the primary limitation of our study. Given that this study was the first application of our PMG prototype for neuromuscular monitoring, we initially designed it as a small-sample preliminary clinical study. And the results from this study are crucial for our subsequent technical optimization. We have added and modified sentences as follows:

(Page10, Line366-371 of PDF version) “In the first place, this study was originally designed as a small-sample observational study for preliminary feasibility validation and subsequent technical optimization of this PMG prototype for neuromuscular monitoring. Consequently, the sample size is insufficient to comprehensively validate the efficacy of this equipment. A more well-designed study with a larger sample size is still necessary to achieve robust results.”

We are very grateful that you could review our manuscript.

Reviewer 2 Report

Comments and Suggestions for Authors

Phonomyography (PMG) offers advantages such as convenience, stability, and the ability to record signals from multiple muscles, making it a promising method for monitoring neuromuscular activity. This study evaluates the effectiveness of a prototype PMG under general anesthesia. A comparison of the effectiveness of the prototype phonomyograph (PMG) and the TOF-Watch SX device in neuromuscular monitoring during surgery under general anesthesia is presented. The study involved 25 adult patients undergoing non-cardiac surgery. Both devices were used to record the response to a single dose of rocuronium. Pearson and Spearman correlation tests revealed a strong correlation between the data obtained from both devices, confirming the reliability of the PMG compared to the established TOF-Watch SX standard in this study. Study highlights the potential of PMG as a reliable method for monitoring neuromuscular blockade in the operating room.

The work is clear for the field and presented in a well-structured manner. The references provided are current and recent, and no excessive self-citation was found. The experimental design is clear, the results are reproducible. Illustrative materials are sufficiently interpretable. The findings are consistent with the results presented.

1. The characteristics and design of the systems from which the signals were recorded are not shown. Algorithms for processing signals and extracting resulting values, such as recovery time, are not described.

2. It is worth noting that all study participants had similar physical status (ASA I-III) and body mass index. This may limit the generalizability of the results to a larger sample of patients. In addition, the PMG method has its limitations and requires further validation and comparison with established methods on a larger volume of data and in a variety of use cases.

3. The text in Figure 1 is difficult to read; the font size needs to be increased. The axis labels on the graphs in Figure 4 are also difficult to read. Check that the links are formatted correctly.

Author Response

Dear reviewer,

Thank you very much for your meticulous review. We have addressed issues regarding the details of the PMG prototype, inclusion and exclusion criteria, and image modifications. We believe that following your suggestions, the quality of our manuscript will be enhanced. The specific details of the revisions are as follows:

Point 1: The characteristics and design of the systems from which the signals were recorded are not shown. Algorithms for processing signals and extracting resulting values, such as recovery time, are not described.

Response 1: Thanks for this insightful comment. Due to the limitation of manuscript space, the characteristics and design of the systems from which the signals were recorded are not shown. We have added these contents as supplementary materials (see Fig. 1S and Fig. 2S), and the algorithms for processing signals and extracting resulting values have been reported in our previous studies [1-3]. Moreover, the PMG prototype (Fig. 2S) has applied for Chinese invention patent, we feel sorry that the policy limits the disclosure of more technical details.

Figure. 1S

Figure. 1S. Schematic representation and validation of the PMG prototype. (a)The illustration of PMG sensing with a conical acoustic chamber in the housing to eliminate ambient noise. (b)The experimental setup to monitor neuromuscular activities using synchronous PMG and surface electromyography (sEMG), and the muscle (Flexor Digitorum Superficialis, FDS) contraction can be selectively induced by electrical stimulation. (c)Time-domain waveform and time-frequency information of PMG and sEMG signals during voluntary muscle contraction [3]. (d)The relation between PMG Root Mean Square (RMS) and the intensity of electrical stimulation without voluntary muscle contraction, the data is from the FDS muscle of a typical subject. It is important to note that sEMG is contaminated by the artifacts of electrical stimulation, thus sEMG is not suitable for the measurement of depth of muscle relaxation throughout the peri-operative period. The phenomena of subgraph (d) indicate that PMG is immune to electrical stimulation artifacts since no PMG response is observed under the influence of low level (i.e., below 10 mA) electrical stimulation, and once the low-frequency vibration of myofiber is evoked by electrical stimulation, the intensity of PMG is positively increased with stimulation.

Figure. 2S

Figure. 2S. Schematic of the PMG prototype.

References

  1. Guo W.C., Fang Y., Sheng X.J. and Zhu X.Y., "Measuring motor unit discharge, myofiber vibration and haemodynamics for enhanced myoelectric gesture recognition." IEEE Transactions on Instrumentation and Measurement, 2023, 72: 4001510.
  2. Guo W.C., Sheng X.J. and Zhu X.Y., "Assessment of muscle fatigue based on motor unit firing, muscular vibration and oxygenation via hybrid mini-grid sEMG, MMG, and NIRS sensing." IEEE Transactions on Instrumentation and Measurement, 2022, 71: 4008010.
  3. Guo W.C., Sheng X.J., Liu H.H. and Zhu X.Y., “Mechanomyography Assisted Myoeletric Sensing for Upper-extremity Prostheses: a Hybrid Approach”, IEEE Sensors Journal, 2017, 17(10):3100-3108.

And to provide readers with more information, we have renewed the supplementary material and added the following sentence:

(Page3, Line120-121 of PDF version) “The characteristics and design of the PMG prototype for signals recording, processing, and extracting could be found in our previous studies [21,22,23].”

Point 2: It is worth noting that all study participants had similar physical status (ASA I-III) and body mass index. This may limit the generalizability of the results to a larger sample of patients. In addition, the PMG method has its limitations and requires further validation and comparison with established methods on a larger volume of data and in a variety of use cases.

Response 2: Thank you for your meticulous comments. To preliminarily demonstrate the effectiveness of this PMG prototype for intraoperative neuromuscular monitoring, we referenced the guideline on good clinical practice of the studies of pharmacodynamics of muscle relaxants (“Fuchs-Buder T, Claudius C, Skovgaard LT, Eriksson LI, Mirakhur RK, Viby-Mogensen J; 8th International Neuromuscular Meeting. Good clinical research practice in pharmacodynamic studies of neuromuscular blocking agents II: the Stockholm revision. Acta Anaesthesiol Scand. 2007 Aug;51(7):789-808. doi: 10.1111/j.1399-6576.2007.01352.x”) and previous studies which compared neuromuscular monitoring methods with each other. Thus, the inclusion criteria was set as the aforementioned physical status and body mass index.

However, this selection bias indeed limits the scope of our study to extend to other populations, such as critically ill patients, obese patients, and elderly patients—who may benefit more from neuromuscular monitoring. We have modified the manuscript and added sentences as follows:

(Page3, Line85-86 of PDF version) “All inclusion and exclusion criteria were determined based on the Good Clinical Practice Guideline and previous studies[11,14].“

(Page10, Line373-377 of PDF version) “In the third place, although the neuromuscular monitoring effects of PMG prototype in different depths of muscle relaxation was discussed, there is a lack of data on the neuromuscular monitoring effects of PMG prototype in different populations such as critically ill, obese, geriatric, or pediatric patients.”

(Page11, Line391-393 of PDF version) “Moreover, we would conduct randomized controlled trials with expanded sample size to investigate the effect of PMG in neuromuscular monitoring under different NMBs, NMBs antagonists, and in diverse patient populations.”

Point 3: The text in Figure 1 is difficult to read; the font size needs to be increased. The axis labels on the graphs in Figure 4 are also difficult to read. Check that the links are formatted correctly.

Response 3: Thank you so much for your careful check. The resolution and size of the font in Figure 1 have been adjusted to improve readability. Similarly, we have enhanced the resolution and quality of Figure 4 with the axis labels been enlarged and units added.

Once again, we are very grateful for the time and effort you spent reviewing our manuscript.

Reviewer 3 Report

Comments and Suggestions for Authors

Subject: Review of Manuscript Submission Bioengineering-2977041

I have carefully reviewed the manuscript titled "The Effect of Phonomyography Prototype for Intraoperative Neuromuscular Monitoring: A Preliminary Study" submitted to Bioengineering for consideration. This is an interesting preliminary study on a novel PMG prototype system for neuromuscular monitoring. Some additional details on the novelty, randomization, clinical relevance of the results, and future plans would enhance the manuscript. Further validation in larger studies and with other muscle groups appears warranted. With some revisions to address the above points, this could make a nice contribution to the literature on neuromuscular monitoring technologies.

1.       The introduction provides good background and motivation for the study. However, it would be helpful if the authors could more clearly state the specific novel contributions of this work compared to previous studies on phonomyography for neuromuscular monitoring. What gaps does this study address that haven't been covered before?

2.       The methodology is described in detail, which is appreciated. One question was there any randomization in the assignment of the PMG prototype vs TOF-Watch SX to patients? Randomization would help reduce any potential bias. Additionally, providing the specific model/version of the PMG prototype used would aid in reproducibility.

3.       The results show no statistically significant differences between the PMG prototype and TOF-Watch SX for the various pharmacodynamic parameters measured, and correlation analyses demonstrate strong agreement. However, the clinical relevance of the differences seen should be discussed more. For example, is a 2.32 minute longer duration of deep neuromuscular block with the PMG clinically meaningful, even if not statistically significant? Some additional discussion here would strengthen the interpretation of the results.

4.       The Bland-Altman plots are a nice way to visualize the agreement between the two devices. However, the axes labels and units are missing from Figure 4, making it difficult to fully interpret. Please add the parameter being compared (e.g. onset time) and the units (e.g. seconds or minutes) to each subplot.

5.       In the discussion, the authors appropriately point out the preliminary nature of this single-center study and the need for larger studies. Given that this is a prototype device, some commentary on plans for further development and testing of the PMG system would be informative. Are there plans to refine the hardware or software based on these initial results?

6.       The comparison to previously published studies on PMG monitoring is appreciated in the discussion. However, the referenced studies used different muscles (corrugator supercilii vs adductor pollicis). Some acknowledgment of how the choice of muscle may impact results, and the generalizability of the present adductor pollicis findings, would be helpful.

Author Response

Dear reviewer,

Thank you very much for your comprehensive comments and suggestions on our manuscript. Your advice on the introduction, methodology, discussion, future perspectives and graph clarity will assist us in improving this manuscript further. The details of the revisions are as follows:

Point 1: The introduction provides good background and motivation for the study. However, it would be helpful if the authors could more clearly state the specific novel contributions of this work compared to previous studies on phonomyography for neuromuscular monitoring. What gaps does this study address that haven't been covered before?

Response 1: We are grateful for the suggestion. To be clearer and in accordance with the reviewer’s concerns, we modified the manuscript clearly. The majorly revised sentences are as follows:

(Page2, Line56-58 of PDF version) “And whether the effect of PMG on neuromuscular monitoring during different blocking degree is comparable to or exceeds that of TOF-Watch SX at adductor pollicis muscle remains elusive.”

(Page2, Line62-67 of PDF version) “The aim of this preliminary study is comparing the effect of the PMG prototype and TOF-Watch SX to reflect the pharmacodynamic property of single bolus rocuronium during general anesthesia, finishing feasibility analysis and effectiveness evaluation for this PMG prototype, and laying a foundation for device update and further study, bridging the aforementioned research gap and advancing the implementation of neuro-muscular monitoring.”

Point 2: The methodology is described in detail, which is appreciated. One question was there any randomization in the assignment of the PMG prototype vs TOF-Watch SX to patients? Randomization would help reduce any potential bias. Additionally, providing the specific model/version of the PMG prototype used would aid in reproducibility.

Response 2: We appreciate your detailed comments and acknowledge the importance of the issue raised regarding randomization. In this prospective preliminary observational study, as both devices measured muscle relaxation degree on the same individual, a randomization method was not employed. To be clearer, we have modified content as below:

(Page2, Line69-70 of PDF version) “This preliminary prospective observational study was approved by the Ethics Committee of West China Hospital of Sichuan University on 8 July 2022 (2022-978).”

Additionally, the results of this study contribute to algorithm improvements and hardware updates for the PMG prototype. Therefore, the current device would differ from those utilized subsequently. Based on your recommendation, we have added a description of the version of this device in the methodology section where the PMG prototype is first mentioned to facilitate the distinction of devices in subsequent studies. (Page3, Line116 of PDF version)

Point 3: The results show no statistically significant differences between the PMG prototype and TOF-Watch SX for the various pharmacodynamic parameters measured, and correlation analyses demonstrate strong agreement. However, the clinical relevance of the differences seen should be discussed more. For example, is a 2.32 minute longer duration of deep neuromuscular block with the PMG clinically meaningful, even if not statistically significant? Some additional discussion here would strengthen the interpretation of the results.

Response 3: Thank you very much for your insights. We strongly agree with your perspective. In scientific research, many results that may not reach statistical significance can still have clinical relevance, thus impacting the decision-making of healthcare professionals and the prognosis of patients. We have added the following discussion to our manuscript:

(Page9-10, Line325-335 of PDF version) “A 2.32 minutes shorter of dNMB period could be explained by the sensitive feature of PMG to detect subtle muscle strength recovery compared with AMG. Nevertheless, data of dNMB duration was limited and exhibited fluctuations with insufficient statistical power. Further large-scale studies focusing on dNMB surgeries are needed to confirm these findings. In any case, this result indicates that PMG could be utilized for dNMB monitoring, filling the research gap of PMG in this degree of muscle relaxation. Furthermore, it suggests the PMG prototype may hold the potential to be a reliable neuromuscular monitoring method for dNMB dependent surgery such as bariatric surgery, aneurysm surgery, and laryngeal microsurgery [40-42]. Yet, its efficacy and whether it could improve clinical outcomes of dNMB still need further investigation.”

(Page10, Line362-365 of PDF version) “Yet, it should be noted that, AMG frequently overestimates the recovery time of neuromuscular blockade. Therefore, the results need to further investigation and validation through a comparison between PMG with other neuromuscular monitoring methods such as electromyography (EMG).”

Point 4: The Bland-Altman plots are a nice way to visualize the agreement between the two devices. However, the axes labels and units are missing from Figure 4, making it difficult to fully interpret. Please add the parameter being compared (e.g. onset time) and the units (e.g. seconds or minutes) to each subplot.

Response 4: Thank you for your rigorous comment. We have improved the resolution of Figure 4 to enhance the clarity of the labels, and we have also added units to the subplots for easier interpretation.

Point 5:  In the discussion, the authors appropriately point out the preliminary nature of this single-center study and the need for larger studies. Given that this is a prototype device, some commentary on plans for further development and testing of the PMG system would be informative. Are there plans to refine the hardware or software based on these initial results?

Response 5: Thank you for your suggestions. There are plans to refine the hardware or software based on these initial results and to conduct further research. We have added a paragraph in the “Discussion” section to the manuscript. The added sentences are as follows:

(Page10-11, Line380-396 of PDF version) “Future works will be focused on enhancing both the PMG prototype itself and clinical trials design. For algorithms, the integration of time domain and frequency domain analysis of PMG, employing both RMS and Power Spectral Density correlation coefficients, is planned to characterize muscle sound properties and to provide individualized neuromuscular monitoring. Additionally, the automatic extraction of spectral energy from PMG signals related to stimulation frequency will be applied to represent intraoperative muscle fiber activity. This feature may provide insights into patients' short-term and long-term prognoses. For hardware, we aim to develop a combined microphone and stimulation electrode to enhance device integration and to create a lighter and more portable PMG device. The exploration of sensor fusion techniques, such as integrating Surface Electromyography (sEMG), to enhance the dimensions and precision of neuro-muscular monitoring would also be initiated. Moreover, we would conduct randomized controlled trials with expanded sample size to investigate the effect of PMG in neuro-muscular monitoring under different NMBs, NMBs antagonists, and in diverse patient populations. Comparative studies will also be conducted between PMG and other quantitative neuromuscular monitoring devices, such as EMG, to further validate the feasibility of PMG for neuromuscular monitoring.”

Point 6: The comparison to previously published studies on PMG monitoring is appreciated in the discussion. However, the referenced studies used different muscles (corrugator supercilii vs adductor pollicis). Some acknowledgment of how the choice of muscle may impact results, and the generalizability of the present adductor pollicis findings, would be helpful.

Response 6: Thank you for your valuable suggestion. In response, we have revised our manuscript as follows:

(Page9, Line313-316 of PDF version) “For now, as the adductor pollicis muscle is recommended for neuromuscular monitoring in clinical guidelines and is the most frequently used site for neuromuscular monitoring in clinical settings results from our study have sufficient universality and promotional value [4].”

(Page10, Line354-362 of PDF version) “Lacking of T90% data and difference in target muscle may be the contrast of results between our study and that of Hemmerling’s [38]. Though previous studies on the corrugator supercilii muscle offered an alternative choice for neuromuscular monitoring. The discrepancy of results between corrugator supercilii muscle and adductor pollicis muscle, probably owing to the motor units and muscle mass characteristics of corrugator supercilii muscle, the suggests that the choice of target muscle can impact the meas-urement of neuromuscular blockade. Given that the adductor pollicis muscle is currently considered the most optimal site for neuromuscular monitoring, we believe results of our study hold certain clinical potential.”

We would like to thank the reviewer again for taking the time to review our manuscript.

Round 2

Reviewer 3 Report

Comments and Suggestions for Authors

I am satisfied with the revision. It can be accepted in its present form.